# CircRNA_012164/MicroRNA-9-5p axis mediates cardiac fibrosis in diabetic cardiomyopathy

**Honglin Wang, Eric Zi Rui Wang, Biao Feng\*, Subrata Chakrabarti\***

Department of Pathology and Laboratory Medicine, University of Western Ontario, London, Ontario, Canada

* subrata.chakrabarti@schulich.uwo.ca (SC); biao.feng@schulich.uwo.ca (BF)

## Abstract

Noncoding RNAs play a part in many chronic diseases and interact with each other to regulate gene expression. MicroRNA-9-5p (miR9) has been thought to be a potential inhibitor of diabetic cardiomyopathy. Here we examined the role of miR9 in regulating cardiac fibrosis in the context of diabetic cardiomyopathy. We further expanded our studies through investigation of a regulatory circularRNA, circRNA_012164, on the action of miR9. We showed at both the *in vivo* and *in vitro* level that glucose induced downregulation of miR9 and upregulation of circRNA_012164 resulted in the subsequent upregulation of downstream fibrotic genes. Further, knockdown of circRNA_012164 shows protective effects in cardiac endothelial cells and reverses increased transcription of genes associated with fibrosis and fibroblast proliferation through a regulatory axis with miR9. This study presents a novel regulatory axis involving noncoding RNA that is evidently important in the development of cardiac fibrosis in diabetic cardiomyopathy.

## 1. Introduction

Diabetes is a rising concern in the world and a major cause of morbidity and mortality for the more than 400 million people living with diabetes [1, 2]. In the next 20 years, the number of people living with diabetes is expected to rise to nearly 700 million, with increasing prevalence of both of type 1 and type 2 diabetes [2]. In either type, the impotence of insulin causes dysfunction in the ability to transport glucose into cells, leading to chronically high blood sugar levels [3, 4]. Chronic diabetic complications are categorized as micro- or macrovascular, affecting capillaries and small vessels or larger vessels respectively [5]. Microvascular complications include diabetic retinopathy, diabetic nephropathy, diabetic neuropathy and diabetic cardiomyopathy (DCM), while macrovascular complications include atherosclerosis and coronary artery disease [6–8].

DCM is the microvascular cardiac complication of diabetes, and it presents with ventricular dysfunction independent of coronary atherosclerosis and hypertension [9–11]. Clinically, DCM is seen with structural and functional deviations, including left ventricular hypertrophy, cardiac fibrosis, and remodeling [12]. Owing to DCM, patients with diabetes without

**Funding:** This work was supported by grants from the Canadian Institutes of Health Research. Grant

was given to Subrata Chakrabarti (no. 173414). The funders had no role in study design, data collection and analysis, decision to publish, or preparation of the manuscript.

**Competing interests:** The authors have declared that no competing interests exist.

hypertension or coronary artery disease suffer higher rates of cardiac dysfunction than non-diabetic cohorts [10]. Microvascular pathology remains the primary contributor of DCM [5, 9, 13, 14]. At the microscopic level, DCM shows interstitial and perivascular fibrosis, cardiomyocyte hypertrophy, and cardiomyocyte death [9, 15–17]. Hyperglycemia leads to activation of deleterious pathways in endothelial cells which promote DCM. Endothelial dysfunction is signified by the prevalence of oxidative stress, an event which occurs in response to hyperglycemic damage [18–20]. In DCM, various pathways are activated in response to oxidative stress, resulting in altered signaling, and increasing the expression of VEGF, TGFβ, and NF-κB, collagen and fibronectin independent of fibroblasts [6, 21–26]. TGF-beta and NF-κB further promote the differentiation of fibroblasts to myofibroblasts [27, 28]. Additionally, endothelial cells can directly contribute to the fibrosis through a process known as endothelial-to-mesenchymal transition (EndMT), whereby the endothelial cells transdifferentiate into mesenchymal cells, contributing to the fibroblast pool [5, 29–33]. Hyperglycemic damage is also able to trigger pro-EndMT signaling [5, 34, 35].

Epigenetic changes are stable and heritable changes to gene expression without a change in the underlying genome [36]. Epigenetic modifications are central to metabolic memory and diabetic complications. Epigenetic modifications include various categories including, histone modification, DNA methylation and alteration of noncoding RNAs (ncRNAs) [37]. ncRNAs RNA molecules which are transcribed but not translated into proteins, and include long noncoding RNA, microRNA (miRNA) and circular RNA (circRNA) [38–41]. miRNA are on average 22 nucleotides long and function through binding to the 3' UTR of target mRNA, leading to degradation in conjunction with a RNA-induced silencing complex [42]. In contrast, circRNA are covalently closed RNA molecules which function as a sponge to sequester miRNA and bind with proteins [43].

MiRNA-9-5p (miR9) is a highly conserved miRNA that has been shown to be involved in a variety of cancers and diabetic complications [44–46]. It is a molecule of interest in the heart with regards to cardiac fibrosis, and is known to be downregulated in diabetes and in response to hyperglycemia [44]. The upstream regulators of miR9 are not fully understood. We hypothesized that downregulation of miR9 in diabetes may be caused, in part, by circRNA-mediated miRNA sponging. A recent study in our lab showed an array of differentially expressed circRNAs in diabetic murine hearts, and we identified several circRNAs with predicted sponging capabilities against miR9 [47]. The current study focuses on one such circRNA, namely mmu_circRNA_012164. We investigated the glucose-induced changes in circRNA_012164 and its ability to influence miR9 in the context of diabetic heart disease.

## 2. Materials and methods

### 2.1 Animals

All animal experiments were performed in accordance with the Canadian Council on Animal Care. All experiments were approved by the University of Western Ontario Council on Animal Care Committee. Both male and female mice were used.

C57BL/6 (B6) mice were randomly divided into diabetic and control groups at 8 weeks of age. The animals were given daily intraperitoneal injections of streptozotocin (STZ) in sodium citrate buffer (pH 4.5, 50 mg/kg) for 5 consecutive days to induce diabetes. Age- and sex-matched littermates were given equal volumes of buffer only injections. Diabetes was confirmed by measuring blood glucose levels (>16.7 mmol/L) following final STZ injection. Following confirmation of diabetes, mice were weighed and checked for blood glucose, urine glucose, and urine ketones weekly. Mice were assessed for cardiac function through

echocardiography. Mice were then sacrificed after 2 months using isoflurane overdose followed by cardiac puncture, and cardiac tissues were collected [44].

**2.1.2 Generation of miR9 transgenic mice.** The mice used were miR9 overexpressing transgenic (m9TG) mice with EC-specific promoter previously generated in our laboratory [44]. The mice had a B6 background. Endothelial cells were further isolated from hearts of M9TG mice and endothelial specificity of miR9 overexpression was confirmed [44].

## 2.2 Cell culture

Mouse Cardiac Endothelial Cells (#CLU510, Cedarlane Laboratories) (MCECs) were used. Cells were incubated in a humidified incubator at 37°C with 5% CO2 in 2% DMEM (Thermo Fisher Scientific Inc) supplemented with 1% amoxicillin (Wisent Inc). The cells were serum-starved for 18 hours, then exposed to normal glucose (5 mmol/L) or high glucose (30 mmol/L) treatments of D-glucose for 48 hours. Mannitol was used as osmotic control.

MCECs were transfected with siRNA targeted circRNA_012164 (50 nM) using Lipofecta-mine 2000 (Invitrogen Canada Inc). Multiple anti-circRNA_012164 siRNAs were tested for efficiency, and one was chosen based on knockdown capacity (S1 Table). Scrambled siRNA was used as a control. SiRNA were designed based on complementarity to the backsplice junction of circRNA_012164. For double transfection experiments, MCECs were transfected simultaneously with a miRNA-9 antagomir [44]. Cells were incubated with OPTI-MEM and Lipofecta-mine 2000 for 6 hours. Cells were then recovered in DMEM with 1% serum for 6 hours, following which, they were serum starved and treated to glucose as previously described [44].

## 2.3 RNA isolation and qRT-PCR analysis

RNA was isolated from mouse heart tissue and MCECs using TRIzol$^{TM}$ reagent (Invitrogen Canada Inc, ON, Canada). qRT-PCR was performed as in previous publications in our lab [44, 45]. Linear mRNA RT was conducted at 25°C for 10 min, 37°C for 2 hr, 85°C for 5 min, and 85°C for 5 min. CircRNA RT was conducted at 25°C for 10 min, 37°C for 1.5 hr, 50°C for 30 min, and 85°C for 5 min. 2 μg of isolated RNA was used for reverse transcription. Resultant cDNA was diluted 1:10 before qPCR analyses. The reaction mixture consisted of SYBR Green Taq Ready Mix (Sigma-Aldrich), forward and reverse primers, cDNA template and H$_2$O. Primer concentrations for PCR were in the 100–200 nM range. mRNA qPCR was conducted for 50 cycles (95°C for 10 s, 55°C for 10 s, 72°C for 15 s, 80°C for 2 s), miRNA qPCR was conducted for 50 cycles (95°C for 15 s, 60°C for 60 s, 70°C for 2 s), and circRNA qPCR was conducted for 50 cycles (95°C for 30 s, 60°C for 61 s, 72°C for 30 s, 75°C for 1 s). The data was normalized to the housekeeping gene *Actb* to account for variances in amount of template and reverse transcriptase efficiencies, the primer sequences are as follows (Table 1). Levels of cir-cRNA_012164 were assessed using divergent primers designed for the backsplice junction (Table 1). Primers were additionally tested for circRNA specificity through assessment following RNAse R treatment. miR9 levels were assessed with TaqMan miRNA assay (Ambion Inc) following miRNA column extraction (Bio Basic) and normalized to U6 snRNA, in accordance with previous protocol [44].

## 2.4 Protein isolation and analysis

The tissues were lysed and ultrasonicated in RIPA buffer (MilliporeSigma) after harvest. Total protein was collected, and the concentration was measured by using BCA kit (Thermo Fisher Scientific Inc). ELISAs for mouse FN1 (Boster Bio) and COL1A1 (Novus Biologicals) were performed [44].

**Table 1. Primer sequences used for qRT-PCR.**

| Gene | Forward 5'-3' | Reverse 5'-3' |
| --- | --- | --- |
| *Actb* | CCTCTATGCCAACACAGTGC | CATCGTACTCCTGCTTGCTG |
| *Acta2* | CTACTGCCGAGCGTGAGATTGT | GTTTCGTGGATGCCCGCTGACT |
| *Col1a1* | CACCCTCAAGAGCCTGAGTC | GTTCGGGCTGATGTACCAGT |
| *Fn1* | CGGTAGGACCTTCTATTCCT | GATACATGACCCCTTCATTG |
| *Fsp1* | AAGTTGCTCATCACCTTCTGG | GTCCACCTTCCACAAATACTC |
| **CircRNA_012164** | CTAAAGCCGTAAAGCCAAAGGC | TCTTGACCTTCTTGGGCTTGG |

## 2.5 Histology

Mouse heart tissues were collected, fixed in 10% formalin, embedded in paraffin, and cut to 5 μm thick sections on positively charged slides. The sections were deparaffinized in xylene and stained with hematoxylin and eosin as well as Masson's trichrome stain [44].

## 2.6 Statistical analysis

The data are expressed as mean ± standard error of the mean. Statistical significances were analyzed by Student's t-test or ANOVA, followed by Tukey test for multiple comparisons. Differences were considered to be statistically significant at values of $p < 0.05$. Statistical analyses and graphs were generated using GraphPad Prism version 8.0.2 for Windows (GraphPad Software).

# 3. Results

## 3.1 circRNA_012164 and microRNA-9-5p levels are altered in the heart in diabetes

Diabetic animals showed evidence of dysmetabolism as expected. For both blood glucose and body weight, diabetic mice of both the B6 and m9TG lines showed hyperglycemia and reduced body weight when compared to non-diabetic (Table 2). They also demonstrated polyuria (not shown). Diabetic m9TG mice showed similar loss of body weight and hyperglycemia as diabetic wildtype mice (Table 2).

To validate previous data from our laboratory, miR9 levels were assessed. We confirmed that miR9 was significantly inhibited by diabetes in the murine heart (Fig 1A). In addition, CircRNA_012164 levels were also elevated in the hearts of diabetic mice (Fig 1B), in accordance with previous literature [47]. Changes in ncRNA expression were consistent across heart tissues of both male and female mice (S1 Fig).

**Table 2. Clinical monitoring of mice at 2 months following streptozotocin treatment.**

| Group | Mean body weight (g) | Mean blood glucose (mmol/L) |
| --- | --- | --- |
| **B6 non-diabetic** | 31.3 ± 3.1 | 9.1 ± 1.2 |
| **B6 diabetic** | 24.2 ± 3.9 | 25.5 ± 1.5 |
| **M9TG non-diabetic** | 33.2 ± 2.7 | 7.9 ± 1.7 |
| **M9TG diabetic** | 25.2 ± 1.8 | 27.0 ± 2.6 |

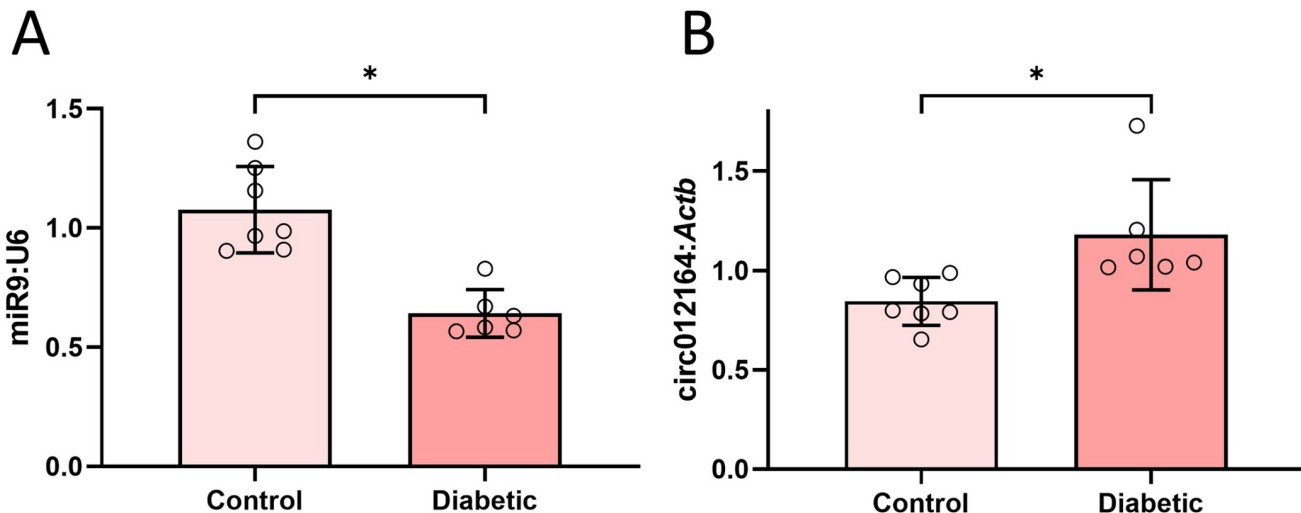

**Fig 1. Diabetes causes alteration in expression of noncoding RNAs.** Assessment of A) microRNA-9 and B) circRNA_01216 levels showed decreased expression of miR-9 and increased expression of circRNA_012164 in male diabetic mice compared to age- and sex-matched non-diabetic controls. Female mice showed similar changes (S1 Fig). (Data are expressed as ± SEM and normalized to U6 or *Actb*. * = significantly different. n = 7/group).

### 3.2 microRNA-9-5p overexpression prevents diabetes-induced cardiac fibrosis

After validating glucose-induced changes in our ncRNAs of interest, we then proceeded to directly examine the role of miR9 in the heart in diabetes. To this extent we used our custom-generated endothelial-specific miR9 overexpressing (m9TG) mice. No phenotypic differences were observed in m9TG mice compared to B6 via echocardiographic analysis, however, B6 diabetic mice showed a pseudonormal E/A ratio when compared to the normal E/A ratios of other groups (Fig 2). The same changes were observed in female mice (S2 Fig).

Gene expression associated with the molecules altered in fibrosis were examined in murine hearts from both B6 and m9TG mice, encompassing both diabetic and non-diabetic conditions. Trichrome staining of murine heart tissue indicated that diabetic B6 mice showed interstitial fibrosis, whereas the age- and sex- matched control diabetic m9TG mice did not (Fig 3A–3D). This analysis further included assessment of mRNAs for *Col1a1* and *Fn1*, genes encoding extracellular matrix proteins, and *S100a4*, a gene that serves as a fibroblast activation marker (Fig 3E–3G). To validate that mRNA expression levels corresponded with observable changes in phenotype, we conducted protein analyses for two key ECM proteins, COL1A1 and FN1 (Fig 3H and 3I). All the genes assessed were significantly elevated in B6 diabetic mouse hearts. These changes indicate diabetes associated cardiac damage. In addition, diabetic animals showed presence of focal myocardial fibrosis. All such alterations were effectively prevented in diabetic m9TG mice (Fig 3).

### 3.3 Knockdown of circRNA_012164 prevented glucose-induced changes in cardiac endothelial cells

After showing the protective effects of miR-9 against DCM-related fibrotic changes *in vivo*, we continued the experiments in isolated mouse cardiac endothelial cells (MCECs) to gain mechanistic insights specific to endothelial cells and endothelial dysfunction. In response to treatment with 30 mM D-glucose, MCECs showed significant elevation in the levels of circRNA_012164 as well as a reduction in miR9 levels corresponding to previous literature

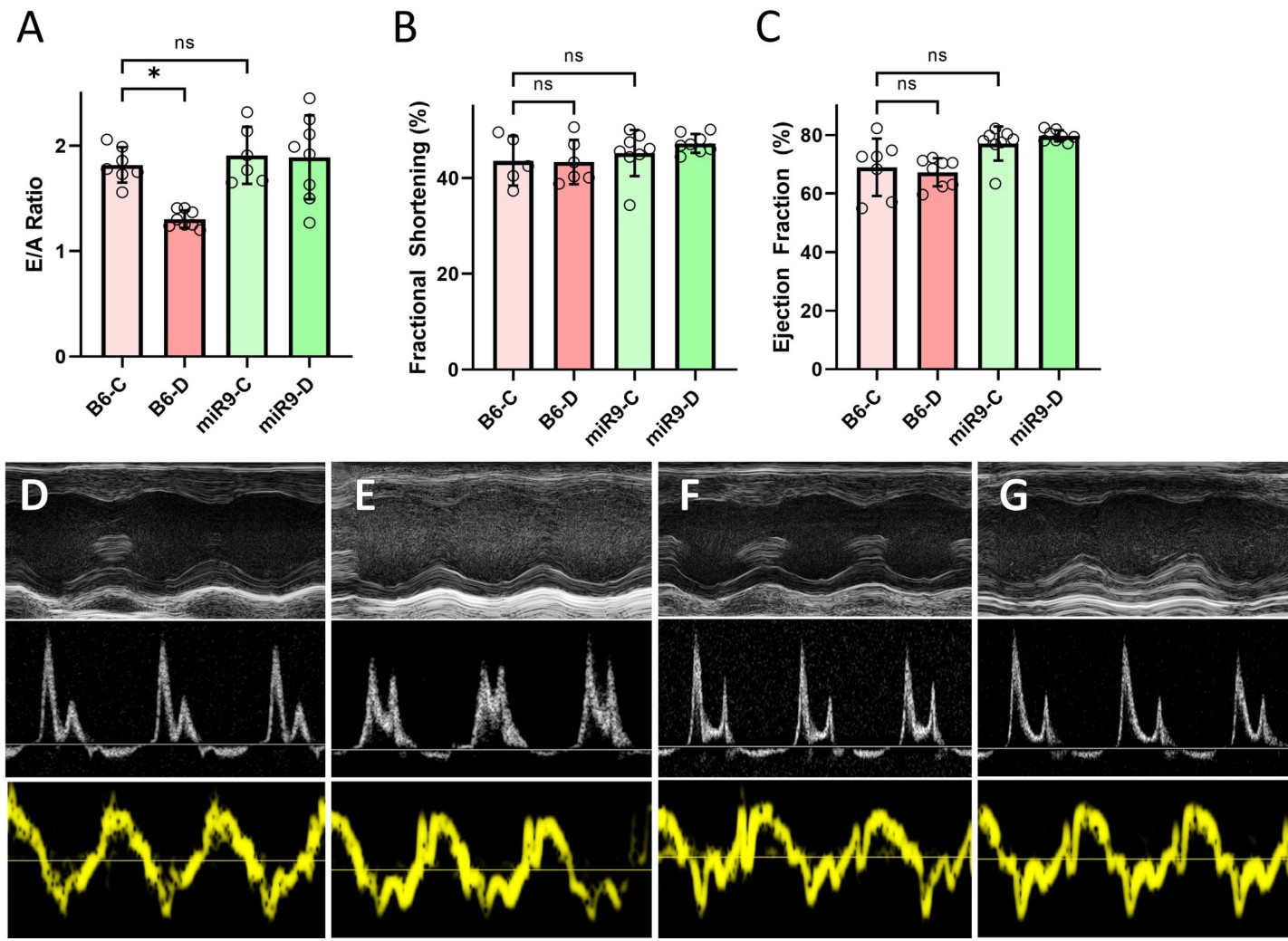

**Fig 2. Endothelial specific microRNA-9 overexpression prevented diabetes induced cardiac abnormalities.** Echocardiographic analysis showed significant reduction of the A) E/A ratio of wild-type diabetic (B6-D) compared to wild-type non-diabetic (B6-C) mice. This reduction was prevented in endothelial specific miR-9 overexpressing transgenic mice. No phenotypic changes were observed in A) E/A ratio, B) fractional shortening, or C) ejection fraction between B6-C and either miR-9 transgenic groups. Representative echocardiography tracings are shown: (top panel) end-systolic diameter and end-diastolic diameter, (middle panel) pulsed wave Doppler recording of mitral inflow velocity, and (bottom panel) tissue Doppler imaging of the LV posterior wall of D) B6-C, E) B6-D, F) miR9-C, and G) miR9-D. Female mice showed similar changes (S2 Fig). (Data are expressed as mean ± SEM. * = significantly different from B6-C. n = 8/group).

and our *in vivo* studies [44] (Fig 4A and 4B). No such changes were observed with osmotic control using mannitol (not shown). Similar to our findings *in vivo*, the mRNA expressions of ECM genes *Col1a1* and *Fn1* were significantly upregulated in response to high glucose (Fig 4D and 4E). Similarly, the levels of *Acta2* and *S100a4*, recognized markers of mesenchymal cells and fibroblasts respectively, also demonstrated an upswing (Fig 4C and 4F). Particularly noteworthy was the heightened expression of *Acta2*, an indication that endothelial cells had initiated the transition toward a mesenchymal phenotype. Knockdown of circRNA_012164 proved to be effective in rescuing aforementioned changes in markers of fibrosis and indicates possible EndMT, while also restoring miR9 levels to those seen in normoglycemic conditions (Fig 4). Interestingly, aside from its protective role in hyperglycemia, circRNA_012164 knockdown had no significant impact on basal levels of any genes, except for *Fn1* (Fig 4E).

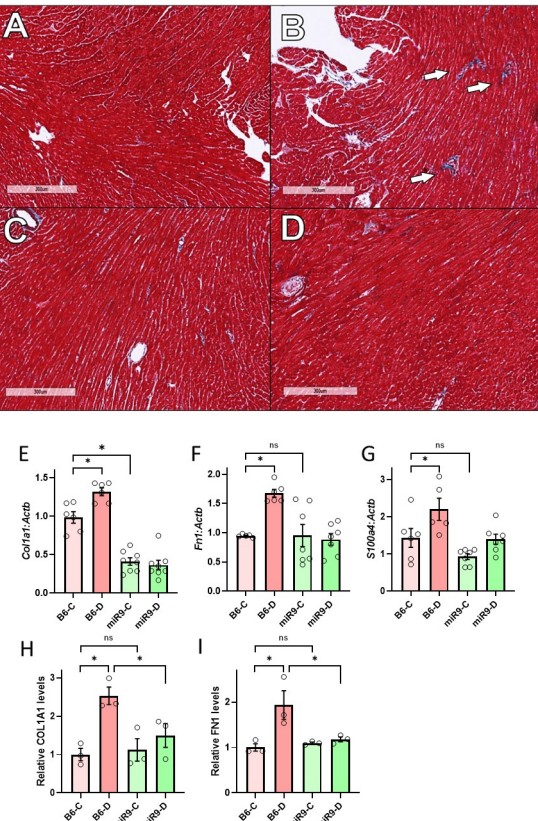

**Fig 3. MicroRNA-9 overexpression prevented diabetes-induced cardiac fibrosis in mice.** Masson's Trichrome staining showed cardiac fibrosis in B) wild-type diabetic (B6-D) compared to A) wild-type non-diabetic (B6-C) hearts, as indicated by arrows. Cardiac fibrosis was prevented in the D) diabetic microRNA-9 overexpressing (endothelial specific) transgenic mice (miR9-D). No differences were observed between A) B6-C and C) miR-9 transgenic non-diabetic (miR9-C) mice. mRNA expressions of E) *Col1a1*, F) *Fn1*, and G) *S100a4* were upregulated in B6-D compared to B6-C hearts. These changes were prevented in miR-9 transgenic mice. mRNA expressions were normalized to *Actb*. Protein levels of H) COL1A1 and I) FN1 were similarly changed in B6-D and miR9-D groups. No differences were observed based on sex differences (S1 Fig). (Data are expressed as mean ± SEM. * = statistically significant. n = 6–8/ group for mRNA analyses, n = 3/group for protein analyses).

## 3.4 Inhibition of microRNA-9 reverts the protective effects of circRNA_012164 knockdown

Having shown the effects of circRNA_012164 knockdown on miR-9 levels and the development of endothelial fibrogenesis and mesenchymal transition, we used a rescue strategy to establish a direct regulatory pathway. To this extent, MCECs were co-transfected with anti-circRNA_012164 siRNA and an antagomir targeting miR9. Co-transfection with both si-circ012164 and miR9 antagomir resulted in a complete nullification of all protective effects of circRNA_012164 knockdown. Co-transfected cells showed significant upregulation of our genes of interest, on par with cells transfected with scrambled oligonucleotides (Fig 5).

## 4. Discussion

The aim of this study was to investigate novel epigenetic mechanisms to develop a better understanding of the pathogenic processes involved in diabetic cardiomyopathy. DCM is an independent risk factor for heart failure in diabetes and manifests clinically as a loss of contractility with cardiac fibrosis and remodeling [16]. Previous research done in our lab has found

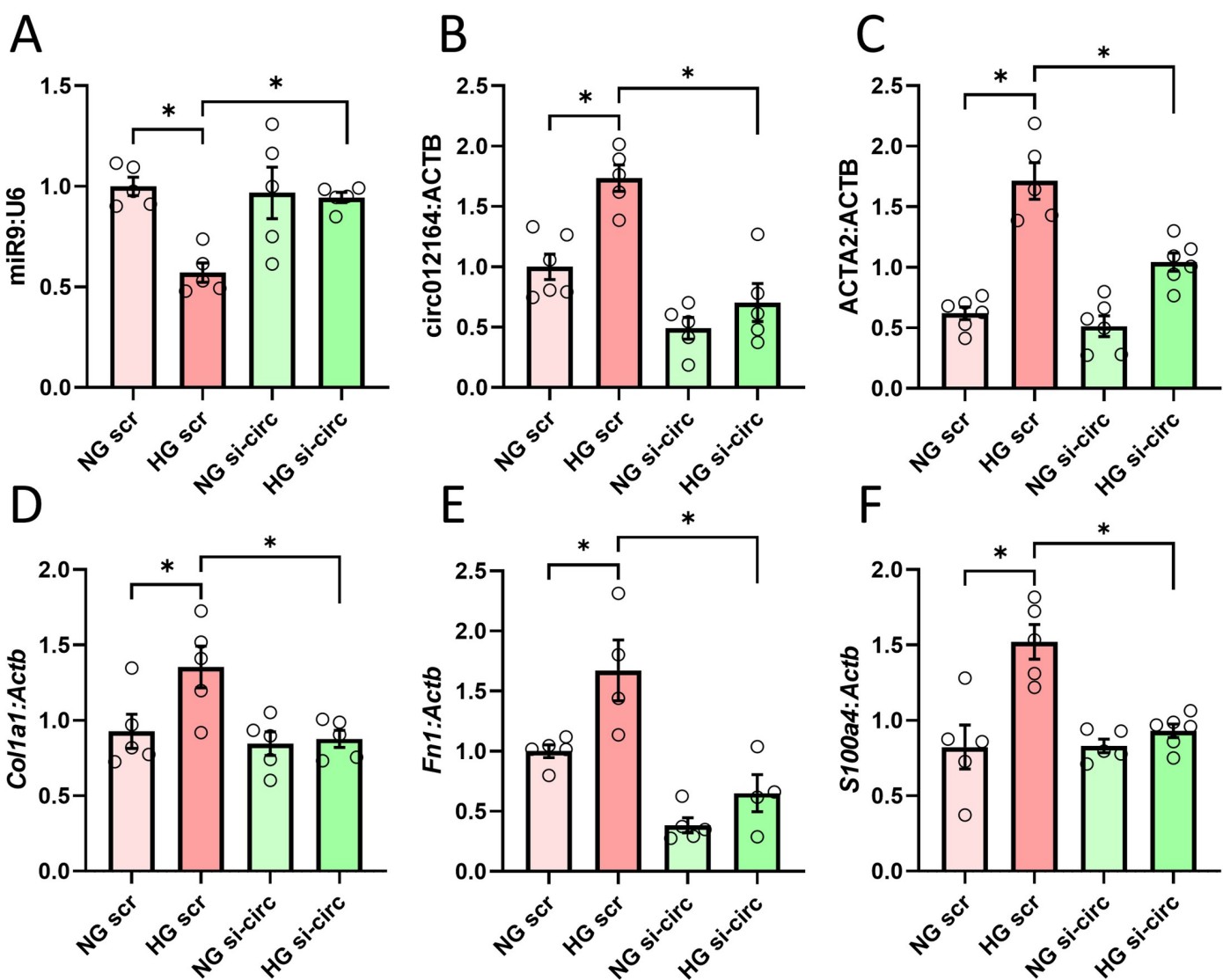

**Fig 4. Knockdown of circRNA_012164 in endothelial cells prevents high glucose induced change in gene expression.** High glucose (30 mM, HG) treatment of mouse cardiac endothelial cells resulted in downregulation of A) miR-9 and upregulation of B) circRNA_012164 along with the downstream genes C) *Acta2* D) *Col1a1*, E) *S100a4*, and F) *Fn1*. siRNA targeted to circRNA_012164 (si-circ) protected cells against HG-induced changes. Data normalized to U6 or *Actb* where applicable. (Data are expressed as mean ± SEM. * = $p < 0.05$, n = 5–6/group.) NG = normal glucose, 5 mM; scr = scrambled oligonucleotides.

the involvement of multiple epigenetic mechanisms, including histone acetylation, long-noncoding RNA, and other miRNAs in the pathogenesis of DCM [44, 48–51]. We had also previously identified several differentially expressed circRNA in the heart in diabetes and their predicted miRNA targets [47].

In this study, we focused on circRNA_012164, which was upregulated in the heart in diabetes. CircRNA_012164 was predicted to regulate miR9, which has previously been identified as a regulator of diabetic complications in multiple organs, including DCM [47]. Our investigation established regulatory pathways in various models, both in *in vitro* and *in vivo*, using both genetically modified and chronically diabetic mice. Overall, our results indicate the existence of the circRNA_012164/miR9 regulatory axis in the pathogenesis of cardiac fibrosis in DCM. Cardiac fibrosis manifests as increased production of ECM proteins such as collagen type I

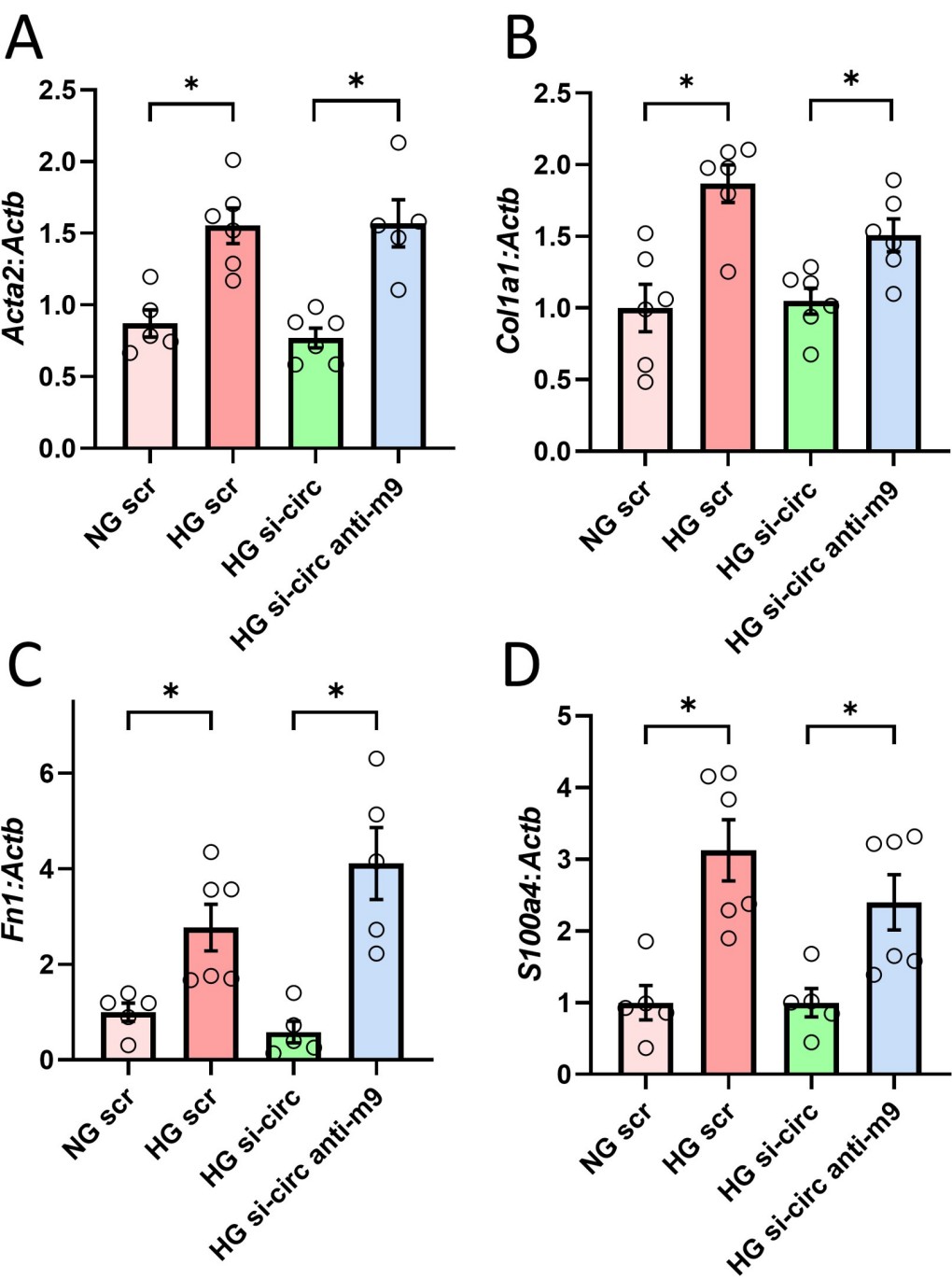

**Fig 5. Knockdown of miR-9 negates the protective effects of circRNA_012164 knockdown in cardiac endothelial cells.** High glucose (30 mM, HG) treatment of mouse cardiac endothelial cells resulted in upregulation of A) *Acta2* B) *Col1a1*, C) *S100a4*, and D) *Fn1*. Knockdown of circRNA_012164 (si-circ) prevented (HG) induced gene expression changes. Additionally inhibiting microRNA-9 (anti-m9) removed protective effect of si-circ and restored the HG phenotype. Data normalized to *Actb*. (Data are expressed as mean ± SEM. * = $p < 0.05$, n = 5–6/group.) NG = normal glucose, 5 mM; scr = scrambled oligonucleotides.

and fibronectin [52]. ECs are capable of both activating fibroblasts through signaling derangement and differentiating into fibroblasts through EndMT, a process by which transcription derangement leads to ECs taking on a mesenchymal phenotype [30].

Levels of miR9 were found to be downregulated in the heart in diabetes, consistent with previous studies [44]. In addition, an increase in transcripts of genes encoding the ECM proteins collagen type 1 (*Col1a1*) and fibronectin (*Fn1*), as well as an increase in the fibroblast marker, *S100a4*. We further show the potency of miR9's regulatory role using an EC-specific miR9 overexpressing transgenic mouse model. Through endothelial-specific expression of miR9, levels of COL1A1 and FN1 were both significantly reduced in diabetes and restored to baseline non-diabetic. In addition, *S100a4* was also significantly reduced and restored to baseline. This is consistent with previous studies that miR9 has an effect on mediating and preventing the development of cardiac fibrosis in response to diabetes [44]. While miR9 has been quantified previously as being a regulator of genes contributing to various diabetic complications through interactions with proteins and other ncRNA, little is known about the mechanisms behind regulation of miR9 [44, 45]. We have previously shown that lncRNA ZFAS1 regulates miR9 through induction of trimethylation of H3K27 via coordination of the polycomb repressive complex 2 [44]. CircRNA_012164 in this case affects miR9 levels at the post-transcriptional level.

CircRNAs are known to regulate miRNA through competitive endogenous binding through sequence complementarity [53]. In diabetic retinopathy, circRNA_0005015 levels were found to be upregulated and facilitated endothelial angiogenesis by promoting EC proliferation, migration and tube formation through inhibition of miR-519d-3p [53]. In DCM, circRNA_010567 was found to be upregulated in diabetic mouse myocardium and in cardiac fibroblasts, and promoted fibrosis through interactions with miR-141 [54]. Our study identified circRNA_012164 as an upstream regulator of miR9. CircRNA_012164 was found to be upregulated in diabetes, consistent with previous findings [47].

Knockdown of circRNA_012164 in normal glucose conditions did not affect baseline expression of most genes, indicating that the effect observed in high glucose conditions could be attributed to glucose-dependent upstream regulators of circRNA_012164. As circRNAs function through sponging, there may be a co-inhibitory effect between circRNA_012164 and miR9. Double knockdown of both circRNA_012164 and miR9 shows that the effect of circRNA_012164 knockdown is tied to miR9. The findings indicate a circRNA_012164/miR9 regulatory axis, and that the profibrotic effect of circRNA_012164 is in part mediated through miR9 inhibition. Regulation of circRNA_012164 and indeed other circRNA have yet to be fully researched.

CircRNAs may be regulated at the transcriptional level through regulation of their parental gene. While the regulators of circRNA_012164 expression are unknown, its parent gene, *Hist1h1c*, codes for the Histone H1.2 protein and we may identify regulators from *Hist1h1c* [47]. *Hist1h1c* has been found to be upregulated and promoting apoptosis in diabetic retinopathy, further suggesting that high glucose may promote expression of *His1h1c* and subsequently, circRNA_012164 [55]. An increasing body of evidence suggests that circRNAs also reciprocally regulates parental gene expression [56, 57]. In this capacity, circRNAs may interface with transcriptional machinery proteins and in the post-transcriptional splicing stage [58, 59]. No evidence exists of this for circRNA_012164, as such this remains an area for further exploration.

Levels of circRNA_012164 changed in whole cardiac tissue as well as in cardiac endothelial cells. This suggests that other cells in the heart may also express circRNA_012164. Previous studies have shown that miR9 is downregulated in cardiomyocytes and in diabetic hearts, and that miR9 inhibits cardiomyocyte death [60]. Should circRNA_012164 be expressed in other cardiac cells as well, it would indicate a potential future exploration into the role of circRNA_012164 in the regulation of cardiomyocytes and resident cardiac fibroblasts. CircRNA_012164 and miR9 may also function in cardiomyocytes and cardiac

fibroblasts through cell-cell interactions with endothelial cells. Gap junctions between endothelial cells and cardiomyocytes and between endothelial cells and cardiac fibroblasts may allow for the transport of ncRNA through connexons [61, 62]. Finally, extracellular vesicles formed from endothelial cells may transport ncRNA to cells without requiring proximity as shown in various studies, they have been indicated as playing a role in endothelial dysfunction as well as other diseases of vascular damage [63, 64]. In diabetes, miR-9-3p has been found to be transferred to retinal endothelial cells in diabetic retinopathy; similar occurrences in DCM require further investigation [65].

In addition, a previous study from our lab have shown that several other circRNA with diabetically altered expression that bind to various miRNA of interest, including miR9 [47]. Other upregulated circRNA, like circRNA_39888, bind to miR-200b, another miRNA of interest in diabetes [47, 50]. Investigation as to the biologic significance of these differentially expressed circRNA may further illustrate a more comprehensive network of interlinked regulatory molecules that contribute to a disease process as complicated as DCM.

In conclusion, we have established the role of circRNA_012164 and its interactions with miR9 in the context of DCM. We have found that circRNA_012164 is upregulated in diabetic murine hearts and that it promotes endothelial dysfunction and fibrogenesis in cardiac endothelial cells. Further investigation into circRNA_012164 upstream regulation as well as other circRNA targeted to miR9 should be done to gain better insight into the role of ncRNA in DCM.

## Supporting information

**S1 Fig. MiR9 and circRNA_012164 expression changes in hearts of diabetic female mice show changes comparable to male mice.**
(TIF)

**S2 Fig.** A) E/A ration, B) fractional shortening percent and C) ejection fraction percent in female mouse echocardiography. No differences were shown between male and female mice.
(TIF)

**S1 Table. Sequence of siRNA targeted to the backsplice junction of circRNA_012164.**
(TIF)

## Author Contributions

**Conceptualization:** Honglin Wang, Biao Feng.

**Data curation:** Honglin Wang.

**Formal analysis:** Honglin Wang.

**Funding acquisition:** Subrata Chakrabarti.

**Investigation:** Honglin Wang.

**Methodology:** Honglin Wang, Biao Feng.

**Project administration:** Honglin Wang, Biao Feng.

**Supervision:** Biao Feng, Subrata Chakrabarti.

**Validation:** Eric Zi Rui Wang.

**Writing – original draft:** Honglin Wang.

**Writing – review & editing:** Honglin Wang, Eric Zi Rui Wang, Subrata Chakrabarti.

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
