## [Decision Letter · Decision Letter 0]

6 May 2024

PONE-D-24-14757CircRNA_012164/MicroRNA-9-5p axis mediates cardiac fibrosis in diabetic cardiomyopathyPLOS ONE

Dear Dr. Chakrabarti, Please submit your revised manuscript by Jun 20 2024 11:59PM. If you will need more time than this to complete your revisions, please reply to this message or contact the journal office at plosone@plos.org. Please include the following items when submitting your revised manuscript:A rebuttal letter that responds to each point raised by the academic editor and reviewer(s). You should upload this letter as a separate file labeled 'Response to Reviewers'.A marked-up copy of your manuscript that highlights changes made to the original version. You should upload this as a separate file labeled 'Revised Manuscript with Track Changes'.An unmarked version of your revised paper without tracked changes. You should upload this as a separate file labeled 'Manuscript'.If applicable, we recommend that you deposit your laboratory protocols in protocols.io to enhance the reproducibility of your results. Protocols.io assigns your protocol its own identifier (DOI) so that it can be cited independently in the future. For instructions see: https://journals.plos.org/plosone/s/submission-guidelines#loc-laboratory-protocols. Additionally, PLOS ONE offers an option for publishing peer-reviewed Lab Protocol articles, which describe protocols hosted on protocols.io. Read more information on sharing protocols at https://plos.org/protocols?utm_medium=editorial-email&utm_source=authorletters&utm_campaign=protocols.

We look forward to receiving your revised manuscript.

Kind regards,

Filomena de Nigris, Ph.D.

Academic Editor

PLOS ONE

Journal Requirements:

"Supported by grants from the Canadian Institutes of Health Research (grant no. 173414 to SC)."

"This work was supported by grants from the Canadian Institutes of Health Research. Grant was given to Subrata Chakrabarti (no. 173414). The funders had no role in study design, data collection and analysis, decision to publish, or preparation of the manuscript."

4. We note that your Data Availability Statement is currently as follows:"All relevant data are within the paper and its Supporting information files."

Additional Editor Comments :

Following reviers comments the decision for manuscript is minor changement

Reviewers' comments:

Reviewer's Responses to Questions

**Comments to the Author**

1. Is the manuscript technically sound, and do the data support the conclusions?

Reviewer #1: Yes

Reviewer #2: Yes

2. Has the statistical analysis been performed appropriately and rigorously? 

Reviewer #1: Yes

Reviewer #2: Yes

3. Have the authors made all data underlying the findings in their manuscript fully available?

Reviewer #1: Yes

Reviewer #2: Yes

4. Is the manuscript presented in an intelligible fashion and written in standard English?

Reviewer #1: Yes

Reviewer #2: Yes

5. Review Comments to the Author

Reviewer #1: Manuscript Number: PONE-D-24-14757

Title: CircRNA_012164/MicroRNA-9-5p axis mediates cardiac fibrosis in diabetic cardiomyopathy

This study elucidated the circRNA_012164 regulatory mechanism of the microRNA-9-5p in the cardiac fibrosis in diabetic cardiomyopathy.

The study presents the results of a research and experiments. The analyses are performed to an acceptable technical standard and are described in sufficient detail.

The manuscript is relatively well written, and the analysis was been appropriately performed.

In discussion, as an example, the authors should add, a description of the mechanism of epigenetic molecules in extracellular vesicles during the vascular remodeling in heart disease (as well as DOI: 10.3390/ijms24087509).

Reviewer #2: The study presents the results of original research. Experiments and statistics are performed to a high technical standard and are described in sufficient detail.Conclusions are supported by the data. tThe authors should report, even in short form, the protocol used for qRT-PCR Analysis specifying the concentrations used.

6. PLOS authors have the option to publish the peer review history of their article (what does this mean?). If published, this will include your full peer review and any attached files.

Reviewer #1: No

Reviewer #2: No

---

## [Author Response · Author response to Decision Letter 0]

21 May 2024

Dear Reviewers,

Thank you for taking the time to review our paper. Your suggestions were taken into consideration and implemented. We hope that these changes are acceptable.

Reviewer 1:

More information on extracellular vesicles has been added. Thank you as well for the provided supportive paper.

Reviewer 2:

Additional information has been provided on qRT-PCR analysis including RNA used and primer concentrations. Further information can be found in previous papers as per our lab’s protocols.

Best Regards,

Subrata Chakrabarti

---

## [Editor Report · Decision Letter 1]

28 May 2024

PONE-D-24-14757R1CircRNA_012164/MicroRNA-9-5p axis mediates cardiac fibrosis in diabetic cardiomyopathyPLOS ONE

Dear Dr. Chakrabarti,

Thank you for submitting your manuscript to PLOS ONE. After careful consideration, we feel that it has merit but does not fully meet PLOS ONE’s publication criteria as it currently stands. Therefore, we invite you to submit a revised version of the manuscript that addresses the points raised during the review process. (Minor points)

We look forward to receiving your revised manuscript.

Kind regards,

Filomena de Nigris, Ph.D.

Academic Editor

PLOS ONE

Journal Requirements:

Additional Editor Comments:

Dear Author,

the reviwers ask for some technical questions regarding PCR condition and concentration

please better specify

---

## [Author Response · Author response to Decision Letter 1]

5 Jun 2024

Dear Reviewer,

Thank you for taking the time to review our paper. Your suggestions were taken into consideration and implemented. We hope that these changes are acceptable.

Additional information has been provided on qRT-PCR analysis including RNA used and primer concentrations. Further information can be found in previous papers as per our lab’s protocols.

Best Regards,

Subrata Chakrabarti

---

## [Decision Letter · Decision Letter 2]

12 Jun 2024

CircRNA_012164/MicroRNA-9-5p axis mediates cardiac fibrosis in diabetic cardiomyopathy

PONE-D-24-14757R2

Dear Dr. Chakrabarti

We’re pleased to inform you that your manuscript has been judged scientifically suitable for publication and will be formally accepted for publication once it meets all outstanding technical requirements.

Kind regards,

Filomena de Nigris, Ph.D.

Academic Editor

PLOS ONE

Reviewers' comments:

Reviewer's Responses to Questions

**Comments to the Author**

1. for reviewer 1 the  authors have adequately addressed comments in a previous round  indicating  accept

for the reviewer  2 tfinal comment is below

Reviewer #2: All comments have been addressed

2. Is the manuscript technically sound, and do the data support the conclusions?

T. 

Reviewer #2: Yes

3. Has the statistical analysis been performed appropriately and rigorously? 

Reviewer #2: Yes

4. Have the authors made all data underlying the findings in their manuscript fully available?

Reviewer #2: Yes

5. Is the manuscript presented in an intelligible fashion and written in standard English?

Reviewer #2: Yes

6. Review Comments to the Author

Reviewer #2: The authors responded exhaustively and completely to all comments, improving the structure of the work.

---

## [Editor Report · Acceptance letter]

18 Jun 2024

PONE-D-24-14757R2 

PLOS ONE

Dear Dr. Chakrabarti, 

I'm pleased to inform you that your manuscript has been deemed suitable for publication in PLOS ONE. Congratulations! Your manuscript is now being handed over to our production team.

Kind regards, 

on behalf of

Prof. Filomena de Nigris 

Academic Editor

PLOS ONE